# Movement patterns of the functional reach test do not reflect physical function in healthy young and older participants

Yoshinao Moriyama[1]*, Takumi Yamada[2], Ryota Shimamura[1], Takehiro Ohmi[3], Masaki Hirosawa[1], Tomoyuki Yamauchi[4], Tomohiro Tazawa[5], Junpei Kato[6]

1 Division of Physical Therapy, Department of Rehabilitation, Tokyo Metropolitan Rehabilitation Hospital, Tokyo, Japan, 2 Department of Physical Therapy, Faculty of Health Sciences, Tokyo Metropolitan University, Tokyo, Japan, 3 Clinical Center for Sports Medicine and Sports Dentistry, Tokyo Medical and Dental University, Tokyo, Japan, 4 Department of Rehabilitation, Nihon University Hospital, Tokyo, Japan, 5 Department of Rehabilitation, Sonodakai Joint Replacement Center Hospital, Tokyo, Japan, 6 Karadacare Business Development Office, NEC Livex, Ltd., Tokyo, Japan

* moriyamayoshinao2020@gmail.com

**Data Availability Statement:** All relevant data are within the paper and its Supporting Information files.

## Abstract

The relationship of the Functional Reach Test (FRT) value with the Center of Pressure Excursion (COPE) and physical function remains unclear, and would be influenced by different population characteristics and movement patterns used in the FRT. Therefore, we explored the relationship between the FRT value and the COPE and physical function in healthy young and older individuals classified according to movement patterns. In 21 healthy young participants (42 sides) and 20 older participants (40 sides), three-dimensional motion analysis was performed during the FRT and physical function assessments. The participants were assigned to two clusters after performing a motion analysis during the FRT. Kinematic and kinetic parameters during the FRT and physical function assessment results were compared between the clusters for both groups. Correlation analysis was performed to investigate the relationships of the FRT value with COPE and physical function parameters in each cluster, in young and older individuals separately. The results showed that the hip strategies could be divided into two groups according to the degree of use (Small Hip Strategy, SHS Group; Large Hip Strategy, LHS Group). In the older SHS group, the FRT values were significantly correlated with the COPE (r = 0.75), toe grip strength (r = 0.62), and the five-times sit-to-stand test time (r = -0.52). In the older LHS group and in both groups of young individuals, there were no significant correlations of the FRT value with any parameters. The FRT value reflects the COPE and physical function only in older individuals using the SHS. This could explain previous discrepant results. As there is no simple relationship between the FRT value and physical function, it is important to include movement strategy assessment when using the FRT in clinical evaluations.

**Funding:** The author(s) received no specific funding for this work.

**Competing interests:** The authors have declared that no competing interests exist.

## Introduction

Older people who have fallen typically limit their activities, which contributes to further deterioration of their physical function [1]. In the guidelines for the Prevention of Falls in Older Persons, balance disorders are considered a contributory factor to falls, and balance assessment and exercises are recommended to prevent falls [2, 3]. Therefore, to relieve the social burdens of falls in older people, it is important to assess, analyze, and understand balance disorders.

Several tests have been introduced in clinical practice to assess balance. The Functional Reach Test (FRT) is one of the most commonly used tests. The FRT was created by Duncan et al. [4]. They defined functional reach as "the maximal distance one can reach forward beyond arm's length, while maintaining a fixed base of support in the standing position" [4]. Duncan et al. [4] and Weiner et al. [5] have reported that increases in age and height result in decreased and increased FRT values, respectively. Concerning the relationship between the FRT value and physical function, a higher FRT value was associated with a faster walking speed and with a longer one-leg-standing time [5].

The FRT was designed to facilitate ease of measurement of the limits of stability (LOS) similar to the center of pressure excursion (COPE) [4]. Alexander defined the LOS as the maximum distance that the center of mass (COM) can be moved safely without requiring change in the base of support [6]. In the horizontal plane, the center of pressure (COP) and the COM are displaced in response to acceleration, while at rest they coincide. Therefore, the COPE has been used to evaluate the LOS. The ability to balance increases with a greater LOS.

The FRT is a useful tool for evaluating LOS without a need for special equipment. However, there is no consensus on the relationship between the FRT value and the COPE. Duncan et al. [4] reported a high correlation (r = 0.71) between these values in a group of male and female individuals aged 20–87 years, while Jonsson et al. [7] and Abiko et al. [8] have reported a low correlation (r = 0.38 and 0.46, respectively) in male and female individuals aged 71.3 ± 4.0 years and in female patients aged 20.1 ± 0.4 years. Furthermore, Maeoka et al. [9] and Wallmann [10] found no correlation in healthy female participants (aged 41.7 ± 11.7 years) as well as in male and female individuals (aged 74.9 ± 8.6 years). A study by Mitani et al. [11] found no correlation in young adults (aged 22.5 ± 1.9 years), but found a high correlation (r = 0.70) in middle-aged women (aged 59.8 ± 4.3 years). Additionally, Portnoy et al. [12] found a moderate correlation between the FRT value and the COPE in a study of healthy young (aged 25.4 ± 1.1 years) and older people (aged 64.5 ± 3.5 years) and in patients with hemiplegic stroke (aged 61.4 ± 10.1, r = 0.482) [12]. These different findings might be attributed to the different population attributes, measurement methods, and movement strategies across studies.

Concerning the measurement methods, Duncan et al. analyzed both the front–back and the left–right component of the COPE [4]. However, in other studies, only the front–back component of the COPE was evaluated [7–12].

Another possible factor for the lack of consensus on the relationship between the FRT value and the COPE is the influence of movement patterns. Many previous studies have not specified and analyzed movement patterns during the FRT [4, 5, 7–12]. However, ankle, hip, and trunk movements are the main components of joint movement during the FRT. Therefore, it seems that there are several different strategies for achieving the same FRT value. Jonsson et al. [7] and Maeoka et al. [9] have reported that a greater anterior trunk tilt angle results in a FRT value, and that the anterior trunk tilt angle was related to the FRT value rather than the COPE. Interestingly, the anterior trunk tilt angle is mainly related to hip flexion movement. According to Tsushima et al. [13], using ankle movement during the FRT results in forward movement of the COP, but does not facilitate a large reaching distance. In contrast, using hip

movement in the FRT causes a small forward movement of the COP, while facilitating a large reaching distance [13].

Concerning the studies that performed movement strategies used during the FRT, Takasaki et al. [14] used a video camera, while Waroquier–Leroy et al. [15] and Wernick–Robinson et al. [16] have used a force plate and a three-dimensional motion analysis system. Takasaki et al. [14] and Wernick–Robinson et al. [16] have classified movement strategies during the FRT based on ankle and hip movements, whereas Waroquier–Leroy et al. [15] used cluster analysis to divide two similar groups. These studies demonstrated the importance of the ankle and hip movements, but the relationship between the FRT values and the COPE was not examined.

Similar to the relationship between the FRT values and the COPE, there is no consensus on the relationship between the FRT values and physical function or falls. Weiner et al. [5] and Fujisawa et al. [17] have reported that the FRT values were highly correlated with other physical functions (Weiner: gait speed, r = 0.71; one-legged standing time, r = 0.64; and Fujisawa: gait speed, r = 0.52). Moreover, Thapa et al. [18] reported that the FRT values were weakly correlated with other physical functions (gait speed, r = 0.35; chair stand, r = 0.39). Some studies have found no correlation and were skeptical concerning the relationship between the FRT and physical function [9, 16, 19, 20]. Regarding the relationship between the FRT values and falls, some studies have found a relationship [21–24], while others have not [10, 25–27]. Furthermore, the results of a meta-analysis by Rosa et al. [28] indicated that a history of falls did not affect the FRT values.

Three movement strategies are used to control posture: hip, ankle, and stepping strategies. While the ankle and hip strategies are stereotactic strategies, the hip strategy is used when the sway is faster or greater, or in cases where the support surface is more unstable. The stepping strategy is a strategy of repositioning to a new support surface by stepping when it is difficult to achieve balance using the other two strategies [29]. As the FRT investigates how far one can reach without taking a step, the ankle and hip strategies remain important. In addition, postural control strategies are affected by aging; especially, older and younger people are more likely to flex their and ankles, respectively [30, 31].

Leroy et al. performed a cluster analysis on FRT data and reported the existence of two clusters. The two clusters differed in age, the COPE, and the hip flexion angle. Among those aged <50 years, 16 of 17 belonged to the cluster with little hip flexion and one belonged to the cluster with large hip flexion. Among those aged >75 years, six out of 10 belonged to the cluster with large hip flexion group and four belonged to the cluster with little hip flexion group [15]. This suggests that age affects the movement strategy during the FRT; however, not all older people use a strategy with large hip flexion. In addition, Maranesi et al.'s [32] study on patients with diabetes reported similar rates of hip strategy use during the FRT. Especially, they reported rates of 58.8% and 56.7% in the groups with and without diabetic neuropathy, respectively. Thus, movement strategies are highly individualized, and it is important to evaluate the strategy used by each individual.

A factor in the lack of consensus in the relationship of the FRT value with the COPE, physical function, and falls is the hip flexion pattern without movement of the COPE. However, previous studies have not implemented kinematic and kinetic analyses using a three-dimensional motion analyzer. We hypothesized that FRT values did not reflect the COPE or physical function when hip strategy was extensively used during the FRT, and that it reflected the COPE when the hip strategy was used sparingly. Therefore, based on the aforementioned, our aim was to classify the joint movement strategies used during the FRT, using a three-dimensional motion analysis system, and to explore the relationship between the FRT value, the COPE, and physical function according to the classified pattern. These explorations might show whether,

depending on the movement pattern used, the FRT value reflected the LOS or physical function, and whether it is necessary to assess the individual's movement pattern in the FRT when performing a clinical evaluation.

## Materials and methods

### Design

This was a cross-sectional observational study.

### Participants

Twenty-one healthy young participants (nine male and 12 female individuals; age, 25.62 ± 2.85 years; height, 1.65 ± 0.09 m; body weight, 57.13 ± 8.90 kg) and 20 older participants (seven male and 13 female individuals; age, 73.71 ± 5.88 years; height, 1.58 ± 0.09 m; body weight, 59.64 ± 8.77 kg) were recruited. Measurements were made on the left and right sides of the body; thus, the measurements were conducted on 42 and 40 sides in 21 young and 20 older participants, respectively. Young participants were recruited from among our hospital staff or university students. Older participants were recruited from the community health-promoting program conducted at our university. The inclusion criteria were the ability to walk without using walking aids for at least 30 min and to raise the upper limb by 90˚. The exclusion criteria were orthopedic or neurological disabilities and musculoskeletal pain during daily activities.

### Instrumentation

Whole body movements were recorded using a three-dimensional motion analysis system (Vicon Nexus; Oxford Metrics, London, UK) with 12 infrared cameras with a sampling frequency of 100 Hz. Markers were applied to the whole body with reference to Vicon and previous studies [33]. Markers were applied according to the Gait Full-body Model plug-in in Vicon (https://www.vicon.com/). Additional markers were used for detailed analysis of the feet [33]. Ground reaction forces were recorded on four force plates (Kisler Japan, Tokyo, Japan) with a sampling frequency of 1000 Hz. The ground reaction force in the foot was measured using two adjacent force plates. To calculate the detailed kinetics in the foot, a measurement method was chosen, in which one foot crossed two force plates, as Satoh et al.'s previous study described [33]. A software for interactive musculoskeletal modeling (SIMM; Musculo-Graphics, Santa Rosa, CA, USA) was used to analyze the kinetics and kinematics of the foot and whole body based on three-dimensional and floor reaction-force data.

### Procedure

The FRT and physical function assessments (toe grip strength, TGS; one-leg standing test, OLS; five times sit-to-stand test, FTSST; timed up and go test, TUG; comfortable walking) were measured. The order of the measurements was randomized using a computer-generated random number table.

For the FRT, the participants were placed in a standing position with one foot across two adjacent force plates, feet parallel to each other, and with the feet placed at shoulder width. The participants flexed the arm at the shoulder to 90˚, with the elbow fully extended (Fig 1). Then, they reached forward as far as possible. The movement pattern and speed during the FRT were not specified. After two practices, measurements of the FRT motion analysis were taken three times on the left side and three times on the right side. The order of the left-and right-side measurements was randomized using a random number table. Among the three trials in a

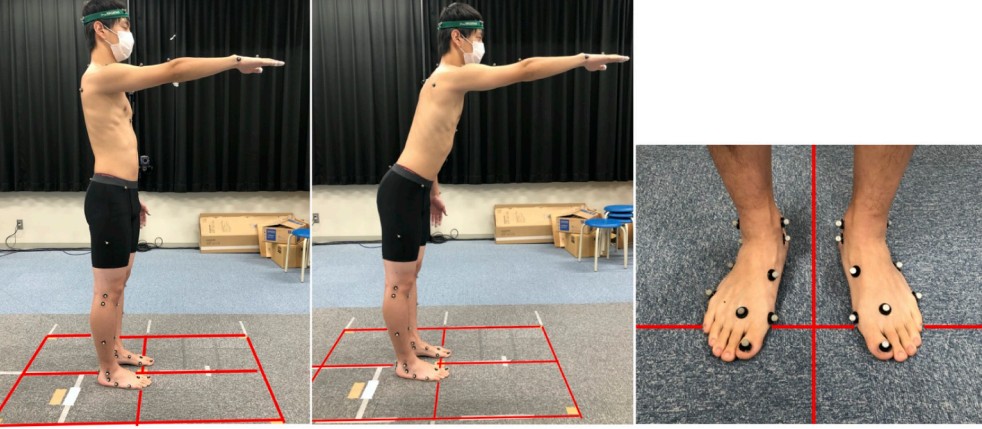

**Fig 1. Measurement of limb position during the Functional Reach Test.**

side, the trial with the maximum FRT values was included in the analysis target in the side. The FRT value in this study was defined from the markers as follows:

FRT value = [fifth metatarsal bone bottom (foot)–third metacarpal head (hand)]–(upper limb length),

where [upper limb length] = [shoulder–third metacarpal head].

= [fifth metatarsal bone bottom (foot)–shoulder].

The TGS was measured using a toe-grip dynamometer (T.K.K. 3360; Takei Co. Ltd., Niigata, Japan). The handle of the force meter was set on the first metatarsophalangeal joint. Measurements were made with the participant sitting with the trunk in a vertical position. The height of the sitting seat and the position of the feet were adjusted so that the hip, knee, and ankle joints were at 90˚. After some practice, the toe-grip strength was measured three times, and the mean value was used in the analysis [34].

We recorded the duration that the participants could stand on one foot in the OLS test. The participants flexed the opposite knee to elevate the foot from the floor and, then, stood as long as possible on the other foot, with the arms hanging down and with the eyes open. The foot, on which they stood, was not allowed to move from the base position, but compensatory movements of the arms and lifted leg were allowed. The duration that the participants could stand on one foot before they touched the floor with the other foot was recorded [35]. After practice, two measurements for each of the left and right sides were made. However, each measurement was limited to a maximum of 60 s. In cases where the maximum period of 60 s was achieved in the first trial, the second trial was not performed.

For the FTSST, the time required for participants to stand up from and sit down on a chair five times, as fast as possible, with their arms folded across the chest was measured. The chair had no armrests and no backrest. The height of the chair's seat was 0.43 m. Measurements were performed in accordance with a previous study [36]. The participants practiced one time and, then, they were measured one time.

For the TUG test, we recorded the time required for participants to stand up from a chair, walk a distance of 3 m, turn, walk back to the chair, and sit down again. The participants were asked to perform their movements as quickly as possible. The chair had no armrests and no backrest. The height of the chair's seat was 0.43 m [37]. The left and right turns were measured one time each. Before the measurements, the participants practiced the left and right turns one time each.

To assess comfortable walking, measurements were performed using a three-dimensional motion analysis device. The participants walked along an 8-m walking path five times. At this time, the participants walked freely, with no restrictions on their steps or rhythm. The average value of the walking speed from five trials was calculated and used as the value of comfortable walking speed.

## Data processing

The missing parts of the marker trajectory data were complemented using the gap-filling function in Vicon. Marker trajectory data were smoothed using a Butterworth filter (cut-off frequency: 6 Hz; filter type, low pass). The ground reaction-force data were smoothed using a Butterworth filter (cut-off frequency: 300 Hz, filter type: low pass). These processes were performed using Vicon. Then, the data processed on Vicon were loaded into the SIMM. Based on these data, the FRT value, the COPE, displacement in each segment, joint angle, and joint moment during the trial of maximum FRT were calculated using the SIMM. In this study, motion analysis was performed only for the FRT, and no detailed motion analysis was performed for the other assessments. The value of each item when the FRT showed a maximum value was analyzed. The FRT values, the COPE, and displacements of each segment were extracted in the anterior–posterior direction, and joint angles and moments were extracted in the sagittal plane. In this study, the COPE was defined as the anterior-posterior component of COM displacement to confirm its relationship with other segments and the FRT values.

The FRT value, COPE, and displacement in each segment were normalized by dividing by the participant's height. The joint moments were normalized by dividing by the participant's weight.

## Statistical analysis

The number of participants required for this study was calculated *a priori* to ensure sufficient statistical power. Power estimates were based on a previous study that investigated the correlation between the FRT and the COPE. This calculation, using G*power 3.1 (University of Dusseldorf, Dusseldorf, Germany), revealed that a sample size of 19 participants would be necessary to achieve a difference with an effect size of 0.60, α-value of 0.05, and 80% power. As we aimed to compare two groups according to their movement patterns, it was predicted that more than 38 participants would be needed.

Older and younger individuals were expected to differ markedly in the FRT values and other items, and pooled analysis might hamper investigations of the effect of different movement patterns. Therefore, we analyzed data of the older and younger individuals separately in this study.

We conducted cluster analysis based on the motion analysis data obtained during the FRT. Moreover, we performed a comparison between the clusters to investigate the influence of different movement patterns, referring to the previous study by Leroy et al [15]. An ascendant hierarchical clustering analysis (using Ward's method, based on Euclidean distances [38] was used to group participants together according to the FRT value, segment displacement, and joint angle during the FRT in each group. Outliers exceeding two standard deviations were replaced by mean values. The participants in whom more than one-third of the data were outliers were excluded from the analysis. For each measurement item, the Shapiro–Wilk test was used to check the normality of data distribution in each group defined by cluster analysis (i.e., Clusters 1 and 2 in the young and Clusters 1 and 2 in the older group). When normal data distribution was observed, Levene's test was performed to check for homogeneity of variance. Then, an unpaired *t*-test was performed for items that showed a normal distribution and

homogeneity of variance, whereas Welch's t-test was performed for items that showed a normal distribution but did not show homogeneity of variance. The Mann–Whitney U-test was performed for items that did not show a normal data distribution.

Correlation analysis was performed to evaluate the relationship between the FRT value and other items (the COPE during the FRT, and the results of the TGS, OLS, FTSST, TUG, and comfortable walking speed tests) in groups separated by cluster analysis. Pearson's correlation coefficient was calculated for items that showed a normal data distribution, and Spearman's rank correlation coefficient was calculated for items that did not show a normal data distribution.

The effect size (r) was determined using EZR (Saitama Medical Center, Jichi Medical University, Saitama, Japan) [39]. All other analyses were performed using SPSS version 25 (IBM SPSS Inc., Armonk, NY, USA). The significance level was set at 5%.

### Ethics

The study protocol adhered to the guidelines of the Declaration of Helsinki and was approved by the Tokyo Research Safety Ethics Committee of Tokyo Metropolitan University (approval no.18080). In addition, the study participants were fully informed of the content and purpose of the research, and the study procedures were carried out after obtaining written informed consent from the participants.

## Results

In total, 21 young participants (42 sides; age, 25.61 ± 2.85 years; height, 1.65 ± 0.09 m; body weight, 57.10 ± 8.90 kg) and 20 older participants (40 sides; age, 73.72 ± 5.88 years; height, 1.58 ± 0.09 m; body weight, 59.64 ± 8.77 kg), respectively, were included in the analysis. Three trials (one in the young and two in the older) were excluded because more than one-third of their parameters were outliers.

### Comparison of clusters divided by movement patterns

The results of the ascendant hierarchical clustering analysis showed two main clusters in each of the young and older groups (Clusters 1 and 2: young, n = 26 and 15, respectively; older, n = 17 and 21, respectively).

In comparing Cluster 1 vs. 2 in young participants (Fig 2, Table 1), Cluster 2 participants had a higher FRT value, a more anterior head and thorax position, and a more posterior pelvic position than Cluster 1 participants. Concerning the joint angles, Cluster 2 participants

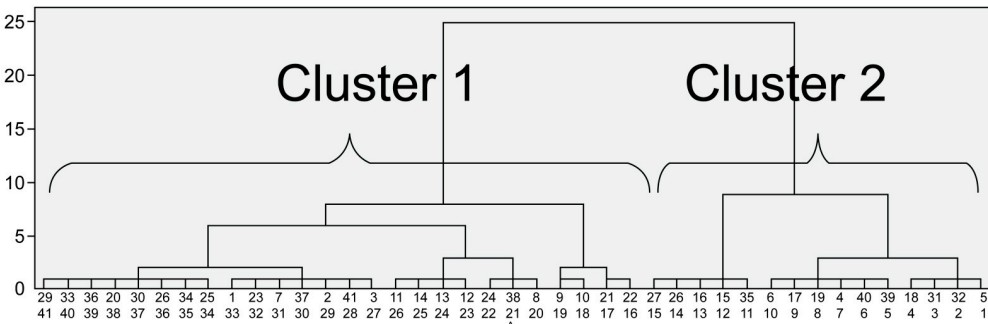

**Fig 2. Dendrogram representing the minimum variance hierarchical classification of the Functional Reach Test patterns in young participants (Ward method; Euclidian distances).**

**Table 1. Comparison of Cluster 1 and Cluster 2 in young participants.**

| | Cluster 1 Mean ± SD | Cluster 2 Mean ± SD | Effect size (r) | 95% CI | p-value |
|---|---|---|---|---|---|
| Anterior Displacement (/height) | | | | | |
| FRT value | 0.22 ± 0.02 | 0.24 ± 0.01 | 0.58 | -0.04– -0.02 | <0.01 |
| Center of Pressure Excursion | 0.05 ± 0.01 | 0.05 ± 0.01 | 0.16 | -0.00–0.01 | 0.32 |
| Head | 0.21 ± 0.03 | 0.27 ± 0.03 | 0.65 | -0.07– -0.03 | <0.01 |
| Thorax | 0.12 ± 0.01 | 0.14 ± 0.01 | 0.53 | -0.02– -0.01 | <0.01 |
| Pelvis | 0.01 ± 0.01 | -0.01 ± 0.01 | 0.66 | 0.01–0.03 | <0.01 |
| Angle (°) | | | | | |
| Lumber flexion | 7.75 ± 4.63 | 13.39 ± 5.55 | 0.49 | -8.91– -2.37 | <0.01 |
| Hip flexion | 26.77 ± 5.88 | 40.20 ± 6.49 | 0.74 | -17.4– -9.42 | <0.01 |
| Knee extension | -6.30 ± 2.65 | -6.46 ± 3.16 | 0.26 | -1.71–2.03 | 0.87 |
| Ankle plantar flexion | 1.11 ± 1.81 | 4.02 ± 2.44 | 0.57 | -4.25– -1.54 | <0.01 |
| Mid-foot dorsal flexion | 6.57 ± 1.73 | 5.67 ± 1.38 | 0.26 | -0.165–1.96 | 0.10 |
| Toe plantar flexion | 5.61 ± 3.31 | 4.74 ± 2.45 | 0.14 | -2.85–1.12 | 0.38 |
| Moment (Nm/kg) | | | | | |
| Lumber extension | 0.84 ± 0.12 | 1.00 ± 0.11 | 0.56 | -0.24– -0.08 | <0.01 |
| Hip extension | 0.58 ± 0.11 | 0.70 ± 0.12 | 0.47 | -0.20– -0.050 | <0.01 |
| Knee flexion | 0.70 ± 0.14 | 0.78 ± 0.11 | 0.34 | -0.14– -0.01 | 0.03 |
| Ankle plantar flexion | 0.66 ± 0.07 | 0.66 ± 0.08 | 0.01 | -0.05–0.05 | 0.95 |
| Mid-foot plantar flexion | 0.42 ± 0.05 | 0.42 ± 0.07 | 0.06 | -0.05–0.03 | 0.72 |
| Toe plantar flexion | 0.14 ± 0.03 | 0.13 ± 0.04 | 0.11 | -0.03–0.01 | 0.48 |
| Physical function | | | | | |
| TGS (/BW) | 0.29 ± 0.09 | 0.32 ± 0.12 | 0.13 | -0.10–0.04 | 0.41 |
| OLS | 60.00 ± 0.00 | 60.00 ± 0.00 | - | - | - |
| FTSST (s) | 6.50 ± 1.53 | 5.90 ± 0.99 | 0.12 | -0.20–1.39 | 0.46 |
| TUG (s) | 5.18 ± 0.54 | 5.16 ± 0.62 | 0.03 | -0.32–0.40 | 0.87 |
| Walking speed (m/s) | 1.33 ± 0.11 | 1.28 ± 0.15 | 0.23 | -0.02–0.15 | 0.15 |

CI, confidence interval; FRT, Functional Reach Test; TGS, toe-grip strength; BW, body weight; OLS, one-leg standing; FTSST, five times sit-to-stand test; TUG, timed up and go test.

showed greater lumbar and hip flexion and greater ankle plantar flexion than Cluster 1 participants. Concerning the joint moments, Cluster 2 participants had greater lumbar and hip extension moments and greater knee flexion moments than Cluster 1 participants. There were no significant differences in physical function assessment results between Cluster 1 and 2 participants.

The toe plantar flexion angle and the FTSST were compared using the Mann–Whitney U test, because of a non-normal data distribution. Other parameters were compared using the unpaired t-test, as the data showed a normal distribution and had homogeneity of variance.

Similarly, after comparing Clusters 1 and 2 in the older group (Fig 3; Table 2), Cluster 2 participants were found to have a higher FRT value, a more anterior head and thorax position, and a more posterior pelvic position than Cluster 1 participants. Concerning the joint angles, Cluster 2 participants had greater hip flexion and greater ankle plantar flexion than Cluster 1 participants. Regarding the joint moments, Cluster 2 participants had greater lumbar and hip extension moments and greater knee flexion moments than those in Cluster 1. There were no significant differences in the results of the physical function assessments in the older group.

The TGS, OLS, and FTSST were compared using the Mann–Whitney U test, because of a non-normal data distribution. The hip flexion angle and mid-foot plantar flexion angle were

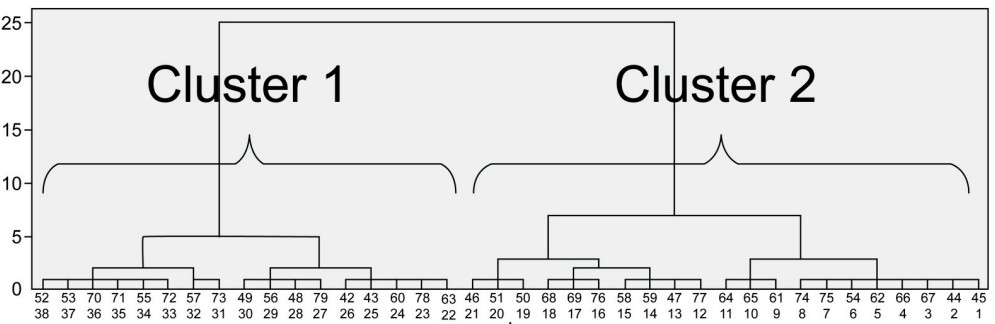

**Fig 3. Dendrogram representing the minimum variance hierarchical classification of the Functional Reach Test patterns in older participants (Ward method; Euclidian distances).**

determined using Welch's t-test because they showed a normal data distribution but did not show homogeneity of variance. Other parameters were compared using the unpaired *t*-test, as the data showed a normal distribution and had homogeneity of variance.

**Table 2. Comparison of Cluster 1 and Cluster 2 in older participants.**

| | Cluster 1 Mean ± SD | Cluster 2 Mean ± SD | Effect size (r) | 95% CI | p-value |
|---|---|---|---|---|---|
| Anterior Displacement (/height) | | | | | |
| FRT value | 0.20 ± 0.03 | 0.24 ± 0.02 | 0.69 | -0.06– -0.26 | <0.01 |
| Center of Pressure Excursion | 0.04 ± 0.01 | 0.05 ± 0.01 | 0.32 | -0.09–0.01 | 0.06 |
| Head | 0.21 ± 0.03 | 0.26 ± 0.03 | 0.65 | -0.08– -0.04 | <0.01 |
| Thorax | 0.11 ± 0.02 | 0.13 ± 0.02 | 0.68 | -0.04– -0.02 | <0.01 |
| Pelvis | -0.01 ± 0.02 | -0.02 ± 0.01 | 0.39 | 0.00–0.02 | 0.02 |
| Angle (°) | | | | | |
| Lumber flexion | 12.00 ± 6.76 | 14.64 ± 5.91 | 0.21 | -6.81–1.53 | 0.21 |
| Hip flexion | 25.45 ± 4.85 | 43.67 ± 7.55 | 0.82 | -22.33– -14.11 | <0.01 |
| Knee extension | -3.53 ± 4.92 | -4.85 ± 5.33 | 0.13 | -4.72–2.10 | 0.74 |
| Ankle plantar flexion | -0.31 ± 2.76 | 3.27 ± 2.38 | 0.58 | -5.28– -1.89 | <0.01 |
| Mid Foot Dorsal flexion | 4.71 ± 0.95 | 4.27 ± 1.76 | 0.15 | -0.48–1.35 | 0.34 |
| Toe plantar flexion | 5.61 ± 3.77 | 3.53 ± 3.17 | 0.29 | -0.21–4.35 | 0.07 |
| Moment (Nm/kg) | | | | | |
| Lumber extension | 0.83 ± 0.14 | 1.01 ± 0.14 | 0.56 | -0.28– -0.10 | <0.01 |
| Hip extension | 0.57 ± 0.16 | 0.73 ± 0.14 | 0.49 | -0.26– -0.06 | <0.01 |
| Knee flexion | 0.58 ± 0.14 | 0.70 ± 0.15 | 0.38 | -0.21– -0.02 | 0.02 |
| Ankle plantar flexion | 0.59 ± 0.11 | 0.59 ± 0.11 | 0.08 | -0.08–0.07 | 0.96 |
| Mid-foot plantar flexion | 0.34 ± 0.10 | 0.36 ± 0.08 | 0.04 | -0.07–0.05 | 0.80 |
| Toe plantar flexion | 0.13 ± 0.03 | 0.12 ± 0.04 | 0.10 | -0.03–0.02 | 0.55 |
| Physical function | | | | | |
| TGS (/BW) | 0.17 ± 0.06 | 0.20 ± 0.10 | 0.01 | -0.08–0.03 | 0.37 |
| OLS (s) | 34.19 ± 22.64 | 41.79 ± 20.91 | 0.18 | -22.0–6.8 | 0.29 |
| FTSST (s) | 6.84 ± 1.74 | 6.54 ± 0.86 | 0.00 | -0.66–1.26 | 0.52 |
| TUG (s) | 5.50 ± 0.57 | 5.25 ± 0.63 | 0.21 | -0.15–0.65 | 0.21 |
| Walking speed (m/s) | 1.44 ± 0.18 | 1.47 ± 0.13 | 0.12 | -0.14–0.07 | 0.47 |

CI, confidence interval; FRT, Functional Reach Test; TGS, toe-grip strength; BW, body weight; OLS, one-leg standing; FTSST, five times sit-to-stand test; TUG, timed up and go test.

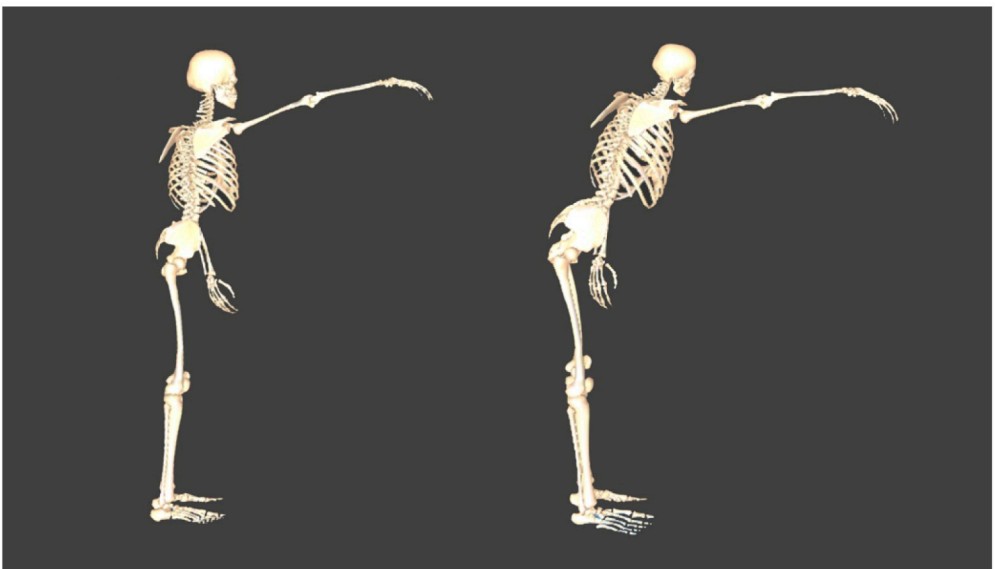

**Fig 4. Typical Functional Reach Test patterns.** Left (Cluster 1): use of the small hip strategy. Right (Cluster 2): use of the large hip strategy.

In terms of kinematics and kinetics, Cluster 2 in both groups appeared to apply the large hip strategy (LHS) during the FRT (Fig 4).

## Correlation between the FRT values and other items in the clusters defined by movement patterns

The results are presented in Table 3. Cluster 1 in the older group (with small hip flexion and plantar flexion) showed significant correlations of the FRT value with the COPE, TGS, and FTSST. In Cluster 2 in the older group (with large hip flexion and plantar flexion), there were no significant correlations of the FRT value with any parameters. In both Clusters 1 and 2 in the young group, there were no significant correlations between the FRT value and the COPE or physical function assessment values.

The TGS in Cluster 2 in the older group, the OLS in Clusters 1 and 2 in the older group, the FTSST in Cluster 1 in the young group and in Cluster 2 in the older group, and walking speed in Cluster 2 in older individuals were calculated using Spearman's rank correlation coefficient,

**Table 3. Coefficients of correlation of parameters with the Functional Reach Test values.**

|  | Young | | Older | |
|---|---|---|---|---|
|  | **Cluster 1** | **Cluster 2** | **Cluster 1** | **Cluster 2** |
| Center of Pressure Excursion (/height) | 0.14 | 0.33 | **0.75**** | 0.32 |
| TGS (/BW) | 0.20 | 0.27 | **0.62**** | 0.01 |
| OLS (s) | - | - | -0.13 | -0.34 |
| FTSST (s) | -0.04 | 0.01 | **-0.52*** | -0.07 |
| TUG (s) | -0.38 | -0.15 | -0.26 | 0.39 |
| Walking speed (m/s) | -0.28 | 0.32 | -0.05 | 0.35 |

FRT, Functional Reach Test; TGS, toe-grip strength; BW, body weight; OLS, one-leg standing; FTSST, five times sit-to-stand test; TUG, timed up and go test

*p <0.05; **p <0.01.

because of a non-normal data distribution. The other parameters were compared using Pearson's correlation coefficient because the data showed a normal distribution.

## Discussion

In this study, we analyzed the FRT movement and measured physical function in both young and older participants to investigate the influence of different movement patterns. Then, we compared movement patterns classified using cluster analysis. The classification results showed differences in the FRT value, hip and ankle joint angles, and posterior pelvic displacement during the FRT. This was similar to the results of a previous study [15]. From the perspective of kinetics, there were differences in the moments at the lumbar, hip, and knee joints, and no differences in the ankle, foot, or toe joints. Accordingly, based on the results of a previous [15] and the present study, we proceeded to analyze the results obtained from two groups, in which a different hip strategy was used in the FRT (Cluster 1, Small hip strategy [SHS]; Cluster 2, LHS). Then, we showed that the FRT values of the older SHS group were correlated with the COPE and physical function assessments, such as the TGS and the FTSST. These findings implied the importance of evaluating movement strategies when performing the FRT as an assessment of physical function.

It has been reported that there was a correlation between the FRT value and height, with a greater FRT value in those with greater height [4, 9, 19]. Therefore, we normalized the FRT value and displacement of a segment, as well as the COPE, by dividing it by the individual's height, to focus on the effect of movement strategies. Similarly, as the joint moment was larger for those with a larger body weight, the moment was normalized by dividing it by the individual's body weight.

Our results of the classification of movement strategies using cluster analysis showed that individuals in the LHS groups had a higher FRT value, a more posterior pelvic displacement, a more anterior head and thoracic displacement, a greater hip flexion angle, and a greater ankle plantar flexion angle than those in the SHS groups. This was similar to the findings of Waroquier–Leroy et al. [15], who reported that posterior pelvic displacement during the FRT was the most distinguishable kinematic parameter. Concerning kinetics, the lumbar and hip extension moments and knee flexion moments were greater in individuals in the LHS groups. This was attributed to the fact that the pelvis was positioned more posteriorly, and the thorax and head were positioned more anteriorly in individuals in the LHS groups, and the moment arms were larger for those joints. There was no difference in the plantar flexion moments of the ankle, foot, and toe joints. This is because there were no differences in the COPE. Posterior pelvic displacement during the FRT is a strategy used to avoid anterior movement of the COP [29, 30]. This is commonly referred to as the hip strategy; when this strategy is used, it increases not only the hip joint moment, but also the lumbar and knee joint moments.

In the young and older groups, individuals in the LHS groups showed a greater FRT value than those in the SHS groups, but there were no differences in physical function between individuals in both groups. This finding supported those of previous studies that found no correlation between the FRT value and physical function parameters [9, 16, 19]. In addition, this finding did not show that hip strategy users had low physical function. Therefore, even if the FRT values are high when using the LHS, this does not necessarily indicate their physical function status. Postural control strategies are affected by aging, and older individuals are more likely to use the hip strategy [30, 31]. However, cluster analysis showed that the hip angle and moment differed significantly between the two clusters in both young and older groups, suggesting that age was not the only determinant of the applied movement strategy during the FRT.

The FRT value did not shown correlations with any parameter in the older LHS group, or in the young SHS or LHS group. In a study by Mitani et al. [11], which examined the relationship between the FRT and COPE in young individuals and middle-aged women, no correlation was found in young people, but a correlation was found in middle-aged women (R = 0.70) [9]. It was proposed that the results might have been influenced by the individuals' height and movement strategies. In this study, we eliminated these influences by normalizing values to height and by dividing the groups based on movement strategies. The lack of correlation, including those with physical function assessments in young individuals, might have been attributed to a ceiling effect.

There was a strong correlation between the FRT value and the COPE (r = 0.75) in the older SHS group, but this correlation was absent in the older LHS group. This suggested that the FRT value could serve as an LOS evaluation in older individuals who used the SHS, but not in those who used the LHS. Moreover, it highlighted the importance of assessing movement patterns when using the FRT to predict the LOS in clinical practice.

In terms of physical function, the FRT values were correlated with the TGS and the FTSST only in the older SHS group. No significant correlations were found for the other physical function items. The strong correlation between the FRT values and the TGS was probably largely caused by using the SHS. Previous studies that used electromyography have reported a shift in muscle activity from ankle plantar flexors to tarsal flexors during reaching, suggesting that toe strength is important when the COM is shifted forward [40]. In previous studies, it was also reported that the TGS and toe compression force were correlated with forward movement of the COP [41, 42]. In the participants of the older SHS group, the FRT values were correlated with the COPE. Thus, the FRT values may have been highly correlated with TGS. In contrast, the young and older LHS groups did not show a correlation between the COPE and FRT values. Consequently, the FRT values did not correlate with TGS. Arai et al. [43] reported that TGS declines with age more easily than other physical functions, such as the FRT, OLS, TUG, knee extension power, and gait velocity. Satoh's study on gait initiation [33] reported that older individuals had smaller ankle, foot, and toe plantar flexion moments than young individuals. In addition, the study showed that the older had smaller plantar flexion moments in the more distal region of the toe [33]. Therefore, older individuals would have low TGS, and when they did not predominantly use the hip strategy during the FRT, poor TGS reflected a low COPE. Thus, the TGS and FRT values would be strongly correlated. In contrast, the TGS value would not be correlated with the FRT values in older individuals using the LHS, as the FRT value would not reflect the COPE.

In this study, we used the FTSST as a standing test, as it is a short and easy test to perform. It is also highly validated with the CS30 and has been reported to be correlated with knee extension muscle strength, which is related to falls [36, 44–47]. Buatois et al. [47] reported a higher risk of falling when the FTSST value was ≥15 s. In this study, the FTSST time for the older group was 6.67 ± 1.30 s (SHS group, 6.84 ± 1.74 s; LHS group, 6.54 ± 0.86 s), and the risk of falling was low. No correlation was found between the FRT and TSST values in the LHS group, while a moderate correlation was found in the SHS group. Schenkman et al. reported two strategies for the sit-to-stand movement (the momentum transfer strategy and the zero-momentum strategy). In the momentum transfer strategy, momentum is gained by using anterior trunk tilt velocity to achieve standing. In this strategy, the point of vertical projection of the COM of the body is behind the COP. In the zero-moment strategy, the torso first leans forward so that the COM is above the feet. Subsequently, the body is placed in a standing position. In this strategy, the body begins to lift from zero velocity (zero momentum), and the projection points of the COM and COP continue to coincide. The momentum transfer strategy is more unstable than the zero-momentum strategy because the COP and COM do not coincide, and it requires coordination of the lower extremities [48]. It has been reported that these strategies

during sit-to-stand movements were affected by speed and that frailer older individuals were more likely to use a strategy, in which the COM and the COP coincide to improve stability [49–52]. In this study, the momentum transfer strategy was more likely to have been used, because of the short FTSST time. Sit-to-stand movement using a momentum transfer strategy might reflect the ability to control the COM and the COP. Therefore, the FTSST might show a moderate correlation with the FRT value in the older SHS group, while the FRT value was strongly correlated with the COPE in these individuals.

Dai et al. [53] devised a modified FRT method, in which the starting posture involves standing with the back and heels against a vertical wall. The wall prohibits backward movement of the pelvis and prohibits the use of the hip strategy. They reported a stronger correlation of the COPE with the modified FRT value (r = 0.82) than with the normal FRT value (r = 0.52) [53]. Our study also showed that the relationship between the FRT value and the COPE changed according to the presence or absence of use of the hip strategy. Therefore, the use of a modified FRT, in which the movement strategies are regulated by a wall, seemed to be a simple and effective method.

Our findings suggested that the FRT has limitations as a method of assessment to reflect physical function. In addition, the effect of movement strategies (mixed use of a hip strategy) could have contributed to the lack of consensus in previous studies on the relationship of the FRT with the COPE and physical function. A modified FRT using a wall on the back that limits the use of hip strategies would be the recommended method of evaluation [53].

## Limitations

The participants in this study were older individuals who participated in a health promotion project several times a week and could walk independently outdoors without a cane. The results of this study's physical function assessment also suggested that they were functional and at a low risk of falling. Therefore, it was unclear whether our findings are generalizable to older individuals with a high risk of falling. In addition, we could not examine the effect of sex and movement strategies on the cutoff value to indicate a risk of falling. In addition, we could not examine the relevance of the movement strategies in other physical function assessments conducted in this study. In the FTSST, we assumed that the duration was short and that a momentum transfer strategy would have been used frequently, but it might have been necessary to analyze the movements of the FTSST and investigate the relation of movement patterns. Future studies should consider these aspects when investigating relationship between fall assessment and movement patterns.

## Conclusion

These results suggested that using the LHS could achieve a high FRT value without greater forward movement of the COP. However, a high FRT value does not necessarily indicate high physical function. Moreover, our findings indicated that the FRT values in individuals using the LHS might not reflect the COPE or their physical function, whereas the FRT values in individuals that use the SHS might reflect the COPE and some physical functions. Therefore, there may not be a simple relationship between the FRT value and physical function. We believe that it is important to include movement strategies when using the FRT for assessment of individuals in clinical practice.

## Supporting information

**S1 File.**
(XLSX)

## Acknowledgments

We would like to thank Editage (www.editage.com) for English language editing.

## Author Contributions

**Conceptualization:** Yoshinao Moriyama, Takumi Yamada, Ryota Shimamura, Takehiro Ohmi, Masaki Hirosawa, Tomoyuki Yamauchi, Tomohiro Tazawa, Junpei Kato.

**Data curation:** Yoshinao Moriyama.

**Formal analysis:** Yoshinao Moriyama.

**Investigation:** Yoshinao Moriyama, Ryota Shimamura, Takehiro Ohmi, Masaki Hirosawa, Tomoyuki Yamauchi, Tomohiro Tazawa, Junpei Kato.

**Methodology:** Yoshinao Moriyama, Takumi Yamada, Ryota Shimamura, Takehiro Ohmi, Masaki Hirosawa, Tomoyuki Yamauchi, Tomohiro Tazawa, Junpei Kato.

**Project administration:** Yoshinao Moriyama.

**Supervision:** Takumi Yamada.

**Visualization:** Yoshinao Moriyama.

**Writing – original draft:** Yoshinao Moriyama.

**Writing – review & editing:** Takumi Yamada, Ryota Shimamura, Takehiro Ohmi, Masaki Hirosawa, Tomoyuki Yamauchi, Tomohiro Tazawa, Junpei Kato.

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
