## [Decision Letter · Decision Letter 0]

3 Jan 2022

PONE-D-21-34488Functional Reach Test with a focus on movement pattern and physical function in healthy young and older participantsPLOS ONE

Dear Dr. Moriyama,

Thank you for submitting your manuscript to PLOS ONE. After careful consideration, we feel that it has merit but does not fully meet PLOS ONE’s publication criteria as it currently stands. Therefore, we invite you to submit a revised version of the manuscript that addresses the points raised during the review process. As you will see, two reviews have been prepared for your manuscript. Both were made by experts in the field. Given that the reviewers have made very different recommendations, I also have read the manuscript and find it really good. Please study both reviewers' comments attentively and make according improvements. Specifically, consider improving the title (cf. reviewer #2). Further, reviewer #1 seems to have had some difficulty to understand some of the aspects, leading to their recommendation of a major. Please do NOT make the changes suggested by this reviewer, but rather try to find ways to make explanations to make these aspects easier to understand. Specifically: possibly you can explain the necessity of using the two clusters such that this is easier to understand. Similarly, the meaning of "sides", which was not clear to reviewer #1. 

We look forward to receiving your revised manuscript.

Kind regards,

Marc H.E. de Lussanet, Ph.D.

Academic Editor

PLOS ONE

Journal Requirements:

2. We note that Figure (1) includes an image of a participant in the study. 

Reviewers' comments:

Reviewer's Responses to Questions

**Comments to the Author**

1. Is the manuscript technically sound, and do the data support the conclusions?

Reviewer #1: Partly

Reviewer #2: Yes

2. Has the statistical analysis been performed appropriately and rigorously? 

Reviewer #1: Yes

Reviewer #2: Yes

3. Have the authors made all data underlying the findings in their manuscript fully available?

Reviewer #1: No

Reviewer #2: Yes

4. Is the manuscript presented in an intelligible fashion and written in standard English?

Reviewer #1: Yes

Reviewer #2: Yes

5. Review Comments to the Author

Reviewer #1: Line 26. It is necessary that the abstract be structured so that each of its parts can be understood.

Line 28-30. The justification for why it is done in two age groups is not clear.

Line 50-137. The introduction is really confusing. Too much data is provided, even a lot that could go into methods or even in discussion. Besides that, it is excessively long, causing the reader to get lost in their reading. It is recommended to redo and simplify it.

Line 138. The justification and the objectives of the work is unclear.

Line 151. I have a question. You have put in the abstract that you analyze the data in two different clusters. However, I do not see that that is contemplated or justified in this section.

Line 154. Perhaps providing the data for the calculation of the number of subjects required is a better fit in the statistical analysis section.

Line 160. What do you mean by 38 sides?

Line 163-5. There must be consistency in the number of decimals of the numbers provided. It is not possible that in some data there are two decimals and in others one decimal.

Line 167-8. This phrase is not understood. Why are 42 sides analyzed in young people and 40 in older people? Wouldn't this phrase be better in the procedure section?

Line 178-9. How were these markers placed?

Line 181-3. It is not clear where it has been described.

Line 188. I don't understand why some of these variables are used. They may be related, but in theory the aim of the study is to correlate FRT with COPE. Please, justify the use of the different variables relating it to what you are trying to demonstrate in the article.

Line 242. The sentence is not clear.

Line 295-6. You should indicate in the statistical analysis of the methods section whether or not the outliers are to be removed.

Line 297. It is necessary to provide information on height, weight, etc. of the participants.

Line 300. It is not clear why he divided the samples into two clusters. What is it based on?

Line 303. For the same reason, it has not been explained in methods whether the results should be compared between clusters or not.

Line 375. The discussion is not clear. The first paragraph does not clearly answer the objectives of the study (they need to be clarified in the introduction). In turn, many of the results already presented in this section are repeated in this section. It would be advisable to redo and synthesize this section.

Line 381. It remains unclear why to split into two clusters.

Line 394. It is recommended not to use the term cluster so much in the discussion, perhaps using what each implies would make it easier to understand this section. You should make it clearer in the methods section what each cluster implies.

Line 434. Although I understand this part of the discussion, having not explained it well in methods, then I do not see sense in this part of the discussion. If different variables are to be considered with respect to the FRT, I would even consider modifying the title to indicate exactly what is being investigated.

487. Perhaps this would go better in limitations.

Figure 2 and 3. Improving the image quality is recommended.

Reviewer #2: This is very well written, and clearly laid out. I could reproduce your study. Your findings are valuable for rehabilitation clinicians and researchers.

My biggest criticism is with the title. it could already indicate the result and highlight this as a must-read for rehabilitation clinicians.

Your abstract and introduction are clear. The abstract has no misleading statements and lays out the study well. The introduction lays out the argument in a logical manner. Lines 65-73 in particular drew me into your subject matter. There was a good scope of literature. It is true that this test and falls prevention have been around for a long time, so there was a wide range of dates, but recent work balanced the older material.

Methodology - please include a 12 months retrospective falls diary for your older participants. Even if it is zero falls, this is important data. It may affect your results if there are falls.

1st trial data on your FRT may have yielded interesting data. If you have saved this, please compare it and consider it for a future paper.

Perhaps in future you could consider movement analysis for FTSST and compare with the movement analysis as described by Sclenkman et al. (future research ideas). You may wish to more clearly outline some of the directions you may want to follow in future research in this topic.

Your recommendation about using a wall behind the subject performing a FRT is strong and valuable to clinicians in the field. Such recommendations are valuable and perhaps could be made a little more visible.

Figures and tables are clear and enhance the paper.

I look forward to seeing this in print.

6. PLOS authors have the option to publish the peer review history of their article (what does this mean?). If published, this will include your full peer review and any attached files.

Reviewer #1: No

Reviewer #2: No

---

## [Author Response · Author response to Decision Letter 0]

29 Jan 2022

Reviewer #1:

The authors would like to thank the reviewer for his/her constructive critique to improve the manuscript. We have made every effort to address the issues raised and to respond to all comments. The revisions are indicated in red font in the revised manuscript. Please, find next a detailed, point-by-point response to the reviewer's comments. We hope that our revisions would meet the reviewer’s expectations.

 Line 26. It is necessary that the abstract be structured so that each of its parts can be understood.

→We would like to thank the reviewer for the suggestion. Please note that we have followed the journal guidelines and formatted the manuscript according to the suggested template. Especially, the template indicates that the Abstract should be presented in an unstructured form. 

Line 28-30. The justification for why it is done in two age groups is not clear.

→Based on previous studies, the lack of consensus could be attributed to the different population attributes, measurement methods, and movement strategies across studies. Therefore, the participants in this study were young and older individuals so as to make a comparison of the findings. This information could not be presented in the Abstract because of the word limit.

Line 50-137. The introduction is really confusing. Too much data is provided, even a lot that could go into methods or even in discussion. Besides that, it is excessively long, causing the reader to get lost in their reading. It is recommended to redo and simplify it.

→We would like to thank the reviewer for the suggestion. Please note that we have made the appropriate revisions based on the reviewer’s suggestion.

Line 138. The justification and the objectives of the work is unclear.

→To address the issue raised by the reviewer, we have added the following part to the revised manuscript:

“Therefore, this study sought to classify the joint movement strategies used during the FRT, using a three-dimensional motion analysis system, and to explore the relationship between the FRT value, the COPE, and physical function according to the classified pattern.” (Lines 145–148)

Line 151. I have a question. You have put in the abstract that you analyze the data in two different clusters. However, I do not see that that is contemplated or justified in this section.

→We would like to thank the reviewer for the comment. Please note that the participants were divided into two clusters as a result of cluster analysis. We have provided this information in the “Statistical analysis” subsection as follows:

“We conducted cluster analysis and comparison between clusters, referring to the previous study by Leroy et al [15].” (Lines 270–271)

 Unfortunately, because of the word limit of the abstract, we could not provide a detailed description of the cluster analysis.

Line 154. Perhaps providing the data for the calculation of the number of subjects required is a better fit in the statistical analysis section.

→We would like to thank the reviewer for the suggestion. However, we believe that the sample size calculation should be presented in the “Design” subsection. 

Line 160. What do you mean by 38 sides?

→We would like to thank the reviewer for the question. To avoid misunderstanding, we have revised this part as follows: 

“As our aim was to compare two groups according to their movement patterns, it was predicted that more than 38 participants would be needed.” (Lines 160–162)

Line 163-5. There must be consistency in the number of decimals of the numbers provided. It is not possible that in some data there are two decimals and in others one decimal.

→The number of decimal places has been changed to two, as per the reviewer’s suggestion.

Line 167-8. This phrase is not understood. Why are 42 sides analyzed in young people and 40 in older people? Wouldn't this phrase be better in the procedure section?

→We would like to thank the reviewer for the questions. To avoid misunderstanding, we have revised this part as follows: 

“Measurements were made on the left and right sides of the body; thus, the measurements were conducted on 42 and 40 sides in 21 young and 20 older participants, respectively.” (Lines 160–162)

Line 177-178. How were these markers placed?

→We would like to thank the reviewer for the question. Please note that we have provided this information as follows:

“Markers were applied to the whole body with reference to VICON and previous studies [33].” (Lines 180–181)

Line 181-3. It is not clear where it has been described.

→We would like to thank the reviewer for the comment. To avoid misunderstanding, we have revised this part as follows: 

“To calculate the detailed kinetics in the foot, a measurement method was chosen, in which one foot crossed two force plates, as Satoh’s previous study described [33].” (Lines 185–187)

Line 188. I don't understand why some of these variables are used. They may be related, but in theory the aim of the study is to correlate FRT with COPE. Please, justify the use of the different variables relating it to what you are trying to demonstrate in the article.

→We would like to thank the reviewer for the comment. Although the measures in this study have been shown to be related to falls, each assesses a different function.

Line 242. The sentence is not clear.

→ We would like to thank the reviewer for the comment. To avoid misunderstanding, we have revised this part as follows: 

“The missing parts of the marker trajectory data were complemented using gap-filling function in Vicon.” (Lines 247–248)

Line 295-6. You should indicate in the statistical analysis of the methods section whether or not the outliers are to be removed.

→As per the reviewer’s insightful suggestion, we have added this information to the revised manuscript as follows:

“We conducted cluster analysis and comparison between clusters, referring to the previous study by Leroy et al [15]. An ascendant hierarchical clustering analysis (using Ward’s method, based on Euclidean distances [38] was used to group participants together according to the FRT value, segment displacement, and joint angle during the FRT in each group. Outliers exceeding two standard deviations were replaced by mean values. The participants in whom more than one-third of the data were outliers were excluded from the analysis.” (Lines 270–276)

Line 297. It is necessary to provide information on height, weight, etc. of the participants.

→As per the reviewer’s insightful suggestion, we have added this information to the revised manuscript as follows:

“In total, 21 young participants (42 sides; age, 25.61 ± 2.85 years; height, 1.65 ± 0.09 m; body weight, 57.10 ± 8.90 kg) and 20 older participants (40 sides; age, 73.72 ± 5.88 years; height, 1.58 ± 0.09 m; body weight, 59.64 ± 8.77 kg), respectively, were included in the analysis.” (Lines 302–305)

Line 300. It is not clear why he divided the samples into two clusters. What is it based on?

Line 303. For the same reason, it has not been explained in methods whether the results should be compared between clusters or not.

→We would like to thank the reviewer for the suggestion. Please note that we have added this information to the revised manuscript as follows:

“We conducted cluster analysis and comparison between clusters, referring to the previous study by Leroy et al [15]. An ascendant hierarchical clustering analysis (using Ward’s method, based on Euclidean distances [38] was used to group participants together according to the FRT value, segment displacement, and joint angle during the FRT in each group. Outliers exceeding two standard deviations were replaced by mean values. The participants in whom more than one-third of the data were outliers were excluded from the analysis.” (Lines 270–276)

＊Because of the sample size, this study analyzed two clusters, similar to a previous study Leroy et al [15].

Line 375. The discussion is not clear. The first paragraph does not clearly answer the objectives of the study (they need to be clarified in the introduction). In turn, many of the results already presented in this section are repeated in this section. It would be advisable to redo and synthesize this section.

→Similar to previous works, we have included a summary of the results at the beginning of the Discussion section. For this reason, we did not consider it to be repetitive.

Line 381. It remains unclear why to split into two clusters.

→Because of the sample size, this study analyzed two clusters as in a previous study by Leroy et al [15]. We have provided this information in the revised manuscript as follows:

“We conducted cluster analysis and comparison between clusters, referring to the previous study by Leroy et al [15].”

(Lines 270–271)

Line 394. It is recommended not to use the term cluster so much in the discussion, perhaps using what each implies would make it easier to understand this section. You should make it clearer in the methods section what each cluster implies.

→The clusters were distinguished based on the results of comparisons between the clusters. Therefore, the description was made in the Discussion section. In addition, when it is necessary to explain the clusters, they are described as "Cluster 1 individuals in the older group (who did not predominantly use the hip strategy)."

Line 434. Although I understand this part of the discussion, having not explained it well in methods, then I do not see sense in this part of the discussion. If different variables are to be considered with respect to the FRT, I would even consider modifying the title to indicate exactly what is being investigated.

→As per the reviewer’s insightful suggestion, we have revised the title as follows:

“Movement patterns of functional reach test not reflecting physical function in healthy young and older participants”

487. Perhaps this would go better in limitations.

→As we were referring to a limitation of the FRT, not to a limitation of this study, this information was added to the Discussion section.

Figure 2 and 3. Improving the image quality is recommended.

→We have improved the quality of these images, as per the reviewer’s suggestion.

Reviewer #2: This is very well written, and clearly laid out. I could reproduce your study. Your findings are valuable for rehabilitation clinicians and researchers. My biggest criticism is with the title. it could already indicate the result and highlight this as a must-read for rehabilitation clinicians.

→The authors would like to thank the reviewer for his/her constructive critique to improve the manuscript. We have made every effort to address the issues raised and to respond to all comments. The revisions are indicated in red font in the revised manuscript. Please, find next a detailed, point-by-point response to the reviewer's comments. We hope that our revisions would meet the reviewer’s expectations.

As per the reviewer’s insightful suggestion, we have revised the title as follows:

“Movement patterns of functional reach test not reflecting physical function in healthy young and older participants”

Your abstract and introduction are clear. The abstract has no misleading statements and lays out the study well. The introduction lays out the argument in a logical manner. Lines 65-73 in particular drew me into your subject matter. There was a good scope of literature. It is true that this test and falls prevention have been around for a long time, so there was a wide range of dates, but recent work balanced the older material.

・Methodology - please include a 12 months retrospective falls diary for your older participants. Even if it is zero falls, this is important data. It may affect your results if there are falls.

→All the elderly in this study could walk without a cane outdoors, and the risk of falling seemed to be low based on the obtained results. However, the presence or absence of falls could not be investigated. In a future study, we will obtain information concerning the history of falls within a 1-year period.

・1st trial data on your FRT may have yielded interesting data. If you have saved this, please compare it and consider it for a future paper.

→We will consider this as an issue in a future research.

・Perhaps in future you could consider movement analysis for FTSST and compare with the movement analysis as described by Sclenkman et al. (future research ideas). You may wish to more clearly outline some of the directions you may want to follow in future research in this topic.

→We would like to thank the reviewer for the suggestion. We will consider this as an issue for future research. Please note that we have discussed this issue as a limitation in the revised manuscript as follows:

“In the FTSST, we assumed that the duration was short and that a momentum transfer strategy would have been used frequently, but it might have been necessary to analyze the movements of the FTSST and investigate the relation of movement patterns. Future studies should consider these aspects when investigating relationship between fall assessment and movement patterns.” (Lines 509–513)

・Your recommendation about using a wall behind the subject performing a FRT is strong and valuable to clinicians in the field. Such recommendations are valuable and perhaps could be made a little more visible.

→Following the reviewer’s suggestion, we have added a part to the Discussion section. The added part is as follows:

“A modified FRT using a wall on the back that limits the use of hip strategies would be the recommended method of evaluation.” (Lines 497–499)

Figures and tables are clear and enhance the paper.

I look forward to seeing this in print.

→We would like to thank the reviewer for the positive comment. We hope that our revisions would meet the reviewer’s expectations and that the revised manuscript is now suitable for publication in your journal.

---

## [Editor Report · Decision Letter 1]

15 Feb 2022

PONE-D-21-34488R1Movement patterns of functional reach test not reflecting physical function in healthy young and older participantsPLOS ONE

Dear Dr. Moriyama,

Thank you for submitting your manuscript to PLOS ONE. After careful consideration, we feel that it has merit but does not fully meet PLOS ONE’s publication criteria as it currently stands. Therefore, we invite you to submit a revised version of the manuscript that addresses the points raised during the review process.

I studied your revised manuscript and decided not to send it to the reviewers again, because the problems are not solved.- The title was changed, but is now grammatically incorrect ("not reflecting" should possibly mean "do not reflect").- You refused to make the necessary changes to the abstract referring to the max number of words. Please note, that the reviewer did not mean that the formatting of the abstract should be changed, but the structure (i.e. the way how the parts and the textual content is structured). There are many guides for how to write an abstract (e.g. doi 10.1007/s13191-013-0299-x; doi: 10.4103/0019-5545.82558). The reviewer is correct that the abstract is rather incomprehensive, and that it is not clear why two age groups are required. - You write, that you made changes to the introduction (Line 50-137), but it cannot be recognized in the marked-up copy which changes you made. Some sentences in the methods are marked in red case, but not in the introduction. Also I noticed that you have made many changes to the text which you have not printed in red case. Please abide by the instructions for the submission of the revision and highlight **all** changes that you made.- I suggest that you revise you revision and you responses to the reviewers' comments and make another submission.

We look forward to receiving your revised manuscript.

Kind regards,

Marc H.E. de Lussanet, Ph.D.

Academic Editor

PLOS ONE
---

## [Author Response · Author response to Decision Letter 1]

4 Mar 2022

Reviewer #1:

The authors would like to thank the reviewer for his/her constructive critique to improve the manuscript. We have made every effort to address the issues raised and to respond to all comments. The revisions are indicated in red font in the revised manuscript. Please, find next a detailed, point-by-point response to the reviewer's comments. We hope that our revisions would meet the reviewer’s expectations.

 Line 26. It is necessary that the abstract be structured so that each of its parts can be understood.

→We would like to thank the reviewer for the suggestion. Please note that we have followed the journal guidelines and formatted the manuscript according to the suggested template. Especially, the template indicates that the Abstract should be presented in an unstructured form. Moreover, please note that we have made revisions in the Abstract to improve the readability.

Line 28-30. The justification for why it is done in two age groups is not clear.

→Based on previous studies, the lack of consensus could be attributed to the different population attributes, measurement methods, and movement strategies across studies. Therefore, the participants in this study were young and older individuals so as to make a comparison of the findings. We have provided this information in the revised manuscript as follows:

“The relationship of the Functional Reach Test (FRT) value with the Center of Pressure Excursion (COPE) and physical function remains unclear, and would be influenced by different population characteristics and movement patterns used in the FRT.” (Lines 26–28)

Line 50-137. The introduction is really confusing. Too much data is provided, even a lot that could go into methods or even in discussion. Besides that, it is excessively long, causing the reader to get lost in their reading. It is recommended to redo and simplify it.

→We would like to thank the reviewer for the suggestion. Please note that we have made the appropriate revisions based on the reviewer’s suggestion.

Line 138. The justification and the objectives of the work is unclear.

→To address the issue raised by the reviewer, we have added the following part to the revised manuscript:

“Therefore, based on the aforementioned, our aim was to classify the joint movement strategies used during the FRT, using a three-dimensional motion analysis system, and to explore the relationship between the FRT value, the COPE, and physical function according to the classified pattern.” (Lines 145–148)

Line 151. I have a question. You have put in the abstract that you analyze the data in two different clusters. However, I do not see that that is contemplated or justified in this section.

→We would like to thank the reviewer for the comment. We have modified this part in the Abstract to avoid confusion. The revised part is as follows:

“The results showed that the hip strategies could be divided into two groups according to the degree of use (Small Hip Strategy, SHS Group; Large Hip Strategy, LHS Group).” (Lines 37–39)

Please note that the participants were divided into two clusters as a result of cluster analysis. We have provided this information in the “Statistical analysis” subsection as follows: 

“We conducted cluster analysis based on the motion analysis data obtained during the FRT. Moreover, we performed a comparison between the clusters to investigate the influence of different movement patterns, referring to the previous study by Leroy et al [15].” (Lines 270–272)

Line 154. Perhaps providing the data for the calculation of the number of subjects required is a better fit in the statistical analysis section.

→We would like to thank the reviewer for the suggestion. Please note that we have moved the part describing the sample size calculation to the “Statistical analysis” subsection, as per the reviewer’s suggestion (Lines 259–265).

Line 160. What do you mean by 38 sides?

→We would like to thank the reviewer for the question. To avoid misunderstanding, we have revised this part as follows: 

“As we aimed to compare two groups according to their movement patterns, it was predicted that more than 38 participants would be needed.” (Lines 264–265)

Line 163-5. There must be consistency in the number of decimals of the numbers provided. It is not possible that in some data there are two decimals and in others one decimal.

→The number of decimal places has been changed to two, as per the reviewer’s suggestion.

Line 167-8. This phrase is not understood. Why are 42 sides analyzed in young people and 40 in older people? Wouldn't this phrase be better in the procedure section?

→We would like to thank the reviewer for the questions. To avoid misunderstanding, we have revised this part as follows: 

“Measurements were made on the left and right sides of the body; thus, the measurements were conducted on 42 and 40 sides in 21 young and 20 older participants, respectively.” (Lines 161–163)

Line 177-178. How were these markers placed?

→We would like to thank the reviewer for the question. Please note that we have provided this information as follows:

“Markers were applied to the whole body with reference to Vicon and previous studies [33].” (Lines 173–174)

Line 181-3. It is not clear where it has been described.

→We would like to thank the reviewer for the comment. To avoid misunderstanding, we have revised this part as follows: 

“To calculate the detailed kinetics in the foot, a measurement method was chosen, in which one foot crossed two force plates, as Satoh et al.’s previous study described.” (Lines 178–180)

Line 188. I don't understand why some of these variables are used. They may be related, but in theory the aim of the study is to correlate FRT with COPE. Please, justify the use of the different variables relating it to what you are trying to demonstrate in the article.

→We would like to thank the reviewer for the comment. Although the measures in this study have been shown to be related to falls, each assesses a different function.

Line 242. The sentence is not clear.

→We would like to thank the reviewer for the comment. To avoid misunderstanding, we have revised this part as follows: 

“The missing parts of the marker trajectory data were complemented using the gap-filling function in Vicon.” (Lines 240–241)

Line 295-6. You should indicate in the statistical analysis of the methods section whether or not the outliers are to be removed.

→As per the reviewer’s insightful suggestion, we have added this information to the revised manuscript as follows:

“We conducted cluster analysis based on the motion analysis data obtained during the FRT. Moreover, we performed a comparison between the clusters to investigate the influence of different movement patterns, referring to the previous study by Leroy et al [15]. An ascendant hierarchical clustering analysis (using Ward’s method, based on Euclidean distances [38] was used to group participants together according to the FRT value, segment displacement, and joint angle during the FRT in each group. Outliers exceeding two standard deviations were replaced by mean values. The participants in whom more than one-third of the data were outliers were excluded from the analysis.” (Lines 270–277)

Line 297. It is necessary to provide information on height, weight, etc. of the participants.

→As per the reviewer’s insightful suggestion, we have added this information to the revised manuscript as follows:

“In total, 21 young participants (42 sides; age, 25.61 ± 2.85 years; height, 1.65 ± 0.09 m; body weight, 57.10 ± 8.90 kg) and 20 older participants (40 sides; age, 73.72 ± 5.88 years; height, 1.58 ± 0.09 m; body weight, 59.64 ± 8.77 kg), respectively, were included in the analysis.” (Lines 303–307)

Line 300. It is not clear why he divided the samples into two clusters. What is it based on?

Line 303. For the same reason, it has not been explained in methods whether the results should be compared between clusters or not.

→We would like to thank the reviewer for the suggestion. Please note that we have added this information to the revised manuscript as follows:

“We conducted cluster analysis based on the motion analysis data obtained during the FRT. Moreover, we performed a comparison between the clusters to investigate the influence of different movement patterns, referring to the previous study by Leroy et al [15]. An ascendant hierarchical clustering analysis (using Ward’s method, based on Euclidean distances [38] was used to group participants together according to the FRT value, segment displacement, and joint angle during the FRT in each group. Outliers exceeding two standard deviations were replaced by mean values. The participants in whom more than one-third of the data were outliers were excluded from the analysis.” (Lines 270–277)

Line 375. The discussion is not clear. The first paragraph does not clearly answer the objectives of the study (they need to be clarified in the introduction). In turn, many of the results already presented in this section are repeated in this section. It would be advisable to redo and synthesize this section.

→We would like to thank the reviewer for the suggestion. Please note that we have added sentence to improve readability:

“In this study, we analyzed the FRT movement and measured physical function in both young and older participants to investigate the influence of different movement patterns.” (Lines 383–384)

Similar to previous works, we have included a summary of the results at the beginning of the Discussion section. For this reason, we did not consider it to be repetitive.

Line 381. It remains unclear why to split into two clusters.

→We would like to thank the reviewer for the suggestion. We have added the following parts to avoid confusion:

“We conducted cluster analysis based on the motion analysis data obtained during the FRT. Moreover, we performed a comparison between the clusters to investigate the influence of different movement patterns, referring to the previous study by Leroy et al [15]. (Lines 270–273)

“In this study, we analyzed the FRT movement and measured physical function in both young and older participants to investigate the influence of different movement patterns.” (Lines 383–384)

“Accordingly, based on the results of a previous [15] and the present study, we proceeded to analyze the results obtained from two groups, in which a different hip strategy was used in the FRT (Cluster 1, Small hip strategy [SHS]; Cluster 2, LHS). Then, we showed that the FRT values of the older SHS group were correlated with the COPE and physical function assessments, such as the TGS and the FTSST.” (Lines 389–394)

Line 394. It is recommended not to use the term cluster so much in the discussion, perhaps using what each implies would make it easier to understand this section. You should make it clearer in the methods section what each cluster implies.

→We would like to thank the reviewer for the suggestion. The clusters were distinguished based on the results of comparisons between the clusters. This information is provided in the Discussion section as follows:

“Accordingly, based on the results of a previous [15] and the present study, we proceeded to analyze the results obtained from two groups, in which a different hip strategy was used in the FRT (Cluster 1, Small hip strategy [SHS]; Cluster 2, LHS). Then, we showed that the FRT values of the older SHS group were correlated with the COPE and physical function assessments, such as the TGS and the FTSST.” (Lines 389–394)

Line 434. Although I understand this part of the discussion, having not explained it well in methods, then I do not see sense in this part of the discussion. If different variables are to be considered with respect to the FRT, I would even consider modifying the title to indicate exactly what is being investigated.

→As per the reviewer’s insightful suggestion, we have revised the title as follows:

“Movement patterns of the functional reach test do not reflect physical function in healthy young and older participants”

487. Perhaps this would go better in limitations.

→As we were referring to a limitation of the FRT, not to a limitation of this study, this information was added to the Discussion section.

Figure 2 and 3. Improving the image quality is recommended.

→We have improved the quality of these images, as per the reviewer’s suggestion.

Reviewer #2: This is very well written, and clearly laid out. I could reproduce your study. Your findings are valuable for rehabilitation clinicians and researchers. My biggest criticism is with the title. it could already indicate the result and highlight this as a must-read for rehabilitation clinicians.

→The authors would like to thank the reviewer for his/her constructive critique to improve the manuscript. We have made every effort to address the issues raised and to respond to all comments. The revisions are indicated in red font in the revised manuscript. Please, find next a detailed, point-by-point response to the reviewer's comments. We hope that our revisions would meet the reviewer’s expectations.

As per the reviewer’s insightful suggestion, we have revised the title as follows:

“Movement patterns of the functional reach test do not reflect physical function in healthy young and older participants”

Your abstract and introduction are clear. The abstract has no misleading statements and lays out the study well. The introduction lays out the argument in a logical manner. Lines 65-73 in particular drew me into your subject matter. There was a good scope of literature. It is true that this test and falls prevention have been around for a long time, so there was a wide range of dates, but recent work balanced the older material.

・Methodology - please include a 12 months retrospective falls diary for your older participants. Even if it is zero falls, this is important data. It may affect your results if there are falls.

→All the elderly in this study could walk without a cane outdoors, and the risk of falling seemed to be low based on the obtained results. However, the presence or absence of falls could not be investigated. In a future study, we will obtain information concerning the history of falls within a 1-year period.

・1st trial data on your FRT may have yielded interesting data. If you have saved this, please compare it and consider it for a future paper.

→We will consider this issue in a future research, as per the reviewer’s suggestion.

・Perhaps in future you could consider movement analysis for FTSST and compare with the movement analysis as described by Sclenkman et al. (future research ideas). You may wish to more clearly outline some of the directions you may want to follow in future research in this topic.

→We would like to thank the reviewer for the suggestion. We will consider this as an issue for future research. Please note that we have discussed this issue as a limitation in the revised manuscript as follows:

“In the FTSST, we assumed that the duration was short and that a momentum transfer strategy would have been used frequently, but it might have been necessary to analyze the movements of the FTSST and investigate the relation of movement patterns. Future studies should consider these aspects when investigating relationship between fall assessment and movement patterns.” (Lines 511–515)

・Your recommendation about using a wall behind the subject performing a FRT is strong and valuable to clinicians in the field. Such recommendations are valuable and perhaps could be made a little more visible.

→Following the reviewer’s suggestion, we have added a part to the Discussion section. The added part is as follows:

“A modified FRT using a wall on the back that limits the use of hip strategies would be the recommended method of evaluation.” (Lines 499–501)

Figures and tables are clear and enhance the paper.

I look forward to seeing this in print.

→We would like to thank the reviewer for the positive comment. We hope that our revisions would meet the reviewer’s expectations and that the revised manuscript is now suitable for publication in your journal.

---

## [Decision Letter · Decision Letter 2]

16 Mar 2022

Movement patterns of the functional reach test do not reflect physical function in healthy young and older participants

PONE-D-21-34488R2

Dear Dr. Moriyama,

We’re pleased to inform you that your manuscript has been judged scientifically suitable for publication and will be formally accepted for publication once it meets all outstanding technical requirements.

Kind regards,

Marc H.E. de Lussanet, Ph.D.

Academic Editor

PLOS ONE

Additional Editor Comments (optional):

Reviewers' comments:

Reviewer's Responses to Questions

**Comments to the Author**

1. If the authors have adequately addressed your comments raised in a previous round of review and you feel that this manuscript is now acceptable for publication, you may indicate that here to bypass the “Comments to the Author” section, enter your conflict of interest statement in the “Confidential to Editor” section, and submit your "Accept" recommendation.

Reviewer #1: All comments have been addressed

2. Is the manuscript technically sound, and do the data support the conclusions?

Reviewer #1: Yes

3. Has the statistical analysis been performed appropriately and rigorously? 

Reviewer #1: Yes

4. Have the authors made all data underlying the findings in their manuscript fully available?

Reviewer #1: Yes

5. Is the manuscript presented in an intelligible fashion and written in standard English?

Reviewer #1: Yes

6. Review Comments to the Author

Reviewer #1: Thank you very much for taking into account all my suggestions.

After the changes made, I think the article has gained a lot in clarity and understandability.

I congratulate you for the work done.

7. PLOS authors have the option to publish the peer review history of their article (what does this mean?). If published, this will include your full peer review and any attached files.

Reviewer #1: No

---

## [Editor Report · Acceptance letter]

23 Mar 2022

PONE-D-21-34488R2 

Movement patterns of the functional reach test do not reflect physical function in healthy young and older participants 

Dear Dr. Moriyama:

I'm pleased to inform you that your manuscript has been deemed suitable for publication in PLOS ONE. Congratulations! Your manuscript is now with our production department. 

Kind regards, 

on behalf of

Dr. Marc H.E. de Lussanet 

Academic Editor

PLOS ONE